# Node Centralities and Classification Performance for Characterizing Node Embedding Algorithms

**Kento Nozawa[1, 2], Masanari Kimura[1], Atsunori Kanemura[2]**
[1]University of Tsukuba, Japan
[2]National Institute of Advanced Industrial Science and Technology (AIST), Japan
{k_nzw,mkimura}@klis.tsukuba.ac.jp, atsu-kan@aist.go.jp

## Abstract

Embedding graph nodes into a vector space can allow the use of machine learning to e.g. predict node classes, but the study of node embedding algorithms is immature compared to the natural language processing field because of a diverse nature of graphs. We examine the performance of node embedding algorithms with respect to graph centrality measures that characterize diverse graphs, through systematic experiments with four node embedding algorithms, four or five graph centralities, and six datasets. Experimental results give insights into the properties of node embedding algorithms, which can be a basis for further research on this topic.

## 1 Introduction

Representation learning for a graph is to assign an embedding vector to each node, so that embeddings can be used as features for machine learning algorithms (Cai et al., 2018). Given a directed or undirected graph $\mathcal{G} = (\mathcal{V}, \mathcal{E})$, a node embedding algorithm finds a mapping from a node $v \in \mathcal{V}$ to a dense and lower-dimensional vector $\mathbf{h} \in \mathbb{R}^d$ ($d \ll |\mathcal{V}|$) called a node embedding. Those embedding vectors are used as feature vectors for e.g. classifying nodes (Perozzi et al., 2014) and predicting links (Grover & Leskovec, 2016); in this way, graph processing tasks such as link prediction are converted to a machine learning problem, often simplifying the entire graph processing procedure.

Many node embedding algorithms have been proposed in the literature, greatly inspired by the influential work in natural language processing (NLP) by Mikolov et al. (2013a), but there is no single algorithm that work better than others on various graphs. Existing algorithms are either based on local information (Tang et al., 2015; Perozzi et al., 2014; Grover & Leskovec, 2016), graphlets (Lyu et al., 2017), or global information (Lai et al., 2017) such as PageRank (Brin & Page, 1998). Graphs have at least two difficulties that do not exist in the NLP context. First, graphs have a large variety e.g. in the edge directionality or their sizes depending on the domain where the data are collected. Second, although it is standard in the NLP field to re-use embeddings obtained from one corpus to other text data (Mikolov et al., 2017), node embeddings cannot be re-used like that because node identity is not maintained for different graphs (word identity is maintained across texts).

In this paper, we examine node embedding algorithms in their node classification performance and analyze them with respect to graph centrality measures such as PageRank. Graph centralities have been employed to characterize various properties of graphs, such as node ranking (Newman, 2010), but we hypothesize they are also useful to characterize node embedding algorithms. Through our systematic experiments with four node embedding algorithms, four or five centrality measures, and six datasets, our findings indicate that an eigenmaps-based algorithm works well for undirected graphs whereas an algorithm with first-order proximity performs well for directed graphs. Our results can be a basis for further research and development on node embedding algorithms.

## 2 Experiment Settings

Our experiment procedure is as follows. Given a graph dataset, we execute a node embedding algorithm and obtain embeddings. Using the embeddings as features, we built a node classifier. Then, for later analysis, we divide nodes into two types: those which classification was correct and

Table 1: Graph datasets for embedding learning and multi-class classification.

| Dataset | Edge | $\lvert \mathcal{V} \rvert$ | $\lvert \mathcal{E} \rvert$ | #Classes |
|---------|------|------|------|----------|
| Cora (Sen et al., 2008) | Directed | 2 708 | 5 429 | 7 |
| PubMed (Sen et al., 2008) | Directed | 19 717 | 44 335 | 3 |
| uCora | Undirected | 2 708 | 5 278 | 7 |
| uPubMed | Undirected | 19 717 | 44 324 | 3 |
| BlogCatalog (Zafarani & Liu, 2009) | Undirected | 10 312 | 333 983 | 39 |
| Flickr (Zafarani & Liu, 2009) | Undirected | 80 513 | 5 899 882 | 195 |

the others. We examine how the distribution of graph centralities are different between the correctly classified and incorrectly classified nodes. Our source code in Docker is publicly available[1].

## 2.1 NODE EMBEDDING ALGORITHMS

We compared the following four node embedding algorithms: Laplacian eigenmaps (Belkin & Niyogi, 2001), LINE-1st and LINE-2nd (Tang et al., 2015), and *node2vec* (Grover & Leskovec, 2016), which are popular and frequently used as the baseline when developing new algorithms.

Laplacian eigenmaps (Belkin & Niyogi, 2001) decompose the normalized Laplacian matrix $L \in \mathbb{R}^{\lvert \mathcal{V} \rvert \times \lvert \mathcal{V} \rvert}$ induced from a graph into a lower-dimensional $\lvert \mathcal{V} \rvert \times d$ matrix by eigendecomposition to map a node to a $d$-dimensional vector ($d \ll \lvert \mathcal{V} \rvert$). Laplacian eigenmaps have two difficulties: 1) they can only work on undirected graphs; 2) they become computationally infeasible for large graphs because of the heavy eigendecomposition.

LINE (Tang et al., 2015) minimizes the Kullback-Leibler divergence between adjacent nodes to learn embeddings. LINE uses only local information (edge connection) and scales to large graphs. LINE has two variants. LINE-1st makes two embeddings $\mathbf{h}_u$ and $\mathbf{h}_v$ similar if nodes $u$ and $v$ are adjacent. LINE-2nd makes these embeddings similar when $u$ and $v$ have many common neighborhood nodes.

The *node2vec* algorithm (Grover & Leskovec, 2016) uses skip-gram with negative sampling (SGNS), originally proposed by Mikolov et al. (2013a;b) for texts. To apply SGNS to a graph, *node2vec* trains skip-gram on node sequences generated by random walks.

Implementation details are found in Appendix A

## 2.2 CENTRALITY MEASURES

We employed the following centralities: degree (for undirected graphs), in-degree and out-degree (for directed graphs), PageRank, closeness, and betweenness. Appendix B describes their definitions.

## 2.3 DATASETS

Table 1 shows six graph datasets we use in this paper; they have a variety in the edge directionality and sizes. Although the original Cora and PubMed datasets are directed, we create their undirected versions, uCora and uPubMed, by ignoring the edge directions, resulting in fewer edges. Cora, PubMed, and their undirected versions are from paper citation networks, and BlogCatalog and Flickr are from social networks. Each node is associated with a class (in Cora and PubMed, seven or three scientific fields, respectively; in BlogCatalog, 39 blog categories; and in Flickr, 195 interest groups).

## 2.4 NODE CLASSIFICATION

To predict node labels from embeddings, we train one-vs-rest logistic regression classifiers with five-fold CV. The regularization parameter for logistic regression is sought from $C \in \{0.25, 0.5, 1, 2, 4\}$ in a nested CV procedure.

---

[1]https://github.com/nzw0301/iclrw2018

Table 2: Micro F1 scores (averaged over five validation folds) of multi-class classification.

| Dataset | Edge | Eigenmaps | LINE-1st | LINE-2nd | *node2vec* |
|---------|------|-----------|----------|----------|----------|
| Cora | Directed | — | **0.805 ± 0.015** | 0.545 ± 0.023 | 0.357 ± 0.005 |
| PubMed | Directed | — | **0.786 ± 0.004** | 0.618 ± 0.011 | 0.531 ± 0.008 |
| uCora | Undirected | **0.861 ± 0.016** | 0.818 ± 0.010 | 0.804 ± 0.014 | 0.837 ± 0.019 |
| uPubMed | Undirected | 0.818 ± 0.003 | 0.791 ± 0.006 | 0.785 ± 0.003 | **0.814 ± 0.009** |
| BlogCatalog | Undirected | **0.390 ± 0.012** | 0.362 ± 0.009 | 0.354 ± 0.007 | 0.348 ± 0.008 |
| Flickr | Undirected | ∗ | **0.363 ± 0.002** | **0.360 ± 0.001** | 0.328 ± 0.001 |

∗ Cannot be computed due to an out-of-memory error on a machine with 128 GB of RAM.

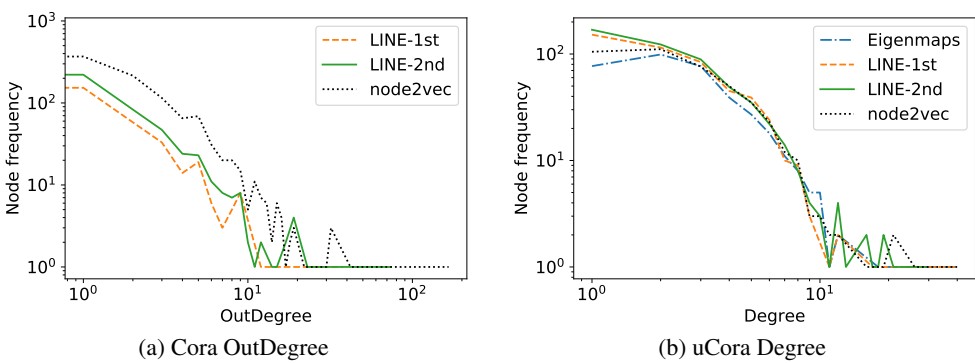

(a) Cora OutDegree          (b) uCora Degree

Figure 1: Power-law plots of graph centrality measures for incorrectly classified nodes.

## 3 RESULTS AND DISCUSSION

Table 2 shows the performance of node classification from node embeddings. In the directed graphs, LINE-1st performed the best, clearly outperforming the other two algorithms. In the undirected graphs, Laplacian eigenmaps obtained the best results almost always, though the difference against *node2vec* was negligible for the uPubMed dataset.

Although the eigenmaps method performed well, it was infeasible to the largest dataset, Flickr, implying the need to make the eigenmaps method scalable. The superior performance of eigenmaps is partly inconsistent with Grover & Leskovec (2016), who claimed eigenmaps are inferior to *node2vec*; the graph is subtle and it is difficult to state something certain.

We found ignoring edge directions can improve the classification performance. Although LINE-1st appears to be the best for the directed graphs, *node2vec* outperformed it on the undirected versions. This implies that embeddings can be refined for classification by converting directed to undirected, and investigating node embedding algorithms for undirected graphs may be more fruitful.

Figure 1 shows power-law plots indicating how the distributions of the degree centralities are different among node embedding algorithms. If the area under a curve is small, then the corresponding algorithm's performance is high since the curve is the frequencies of incorrect classification.

Since the classification performances were largely different across algorithms when classifying Cora nodes (Table 2), it was expected the algorithms obtained different embeddings. This can be justified by Fig. 1a, where the degree centrality curves behave differently for different algorithms, which indicates the classification performance was different for a wide range of degrees.

For uCora, the performance gaps among different algorithms were moderate (Table 2), and in fact Fig. 1b shows that the degree centrality curves were not so discrepant across different algorithms. These curves were most discrepant at the low-degree region; that is, it is suggested that the performance gaps came from the misclassification of low-degree nodes.

We have seen that characterizing node embedding algorithms with respect to their classification performance and node centralities can gain insights into when and how an algorithm works well.

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

## A  ALGORITHM IMPLEMENTATION

We implemented eigenmaps with *scikit-learn*[2] and used the official code of LINE[3] and *node2vec*[4].

The following hyperparameters were selected based on the cross-validation (CV) performance on the node classification task. The embedding dimensionality $d$ was either $128$ or $256$. Whether to normalize or not to normalize embeddings before passing them to a classifier. For *node2vec*, biased random walk parameters $p, q \in \{0.25, 0.5, 1, 2, 4\}$.

We fixed the following hyperparameters based on Tang et al. (2015) and Grover & Leskovec (2016). For the LINE models, we set the number of negative samples to 5, the initial learning rate $\rho$ to $0.025$, and the number of threads to 16. For *node2vec*, the number of random walks per node $r$ to 10, the length of a random walk $l$ to 80, the number of negative sampling to 5, and the number of context nodes to 10.

---

[2]http://scikit-learn.org/stable/modules/manifold.html#spectral-embedding
[3]https://github.com/tangjianpku/LINE
[4]https://github.com/snap-stanford/snap/tree/master/examples/node2vec

# B  NODE CENTRALITY MEASURES

We describe the four node centrality measures used in this study. In the field of network analysis, node centralities are used to characterize nodes depending on their relationship to other nodes (Newman, 2010). Given a graph $\mathcal{G} = (\mathcal{V}, \mathcal{E})$, we denote nodes by $u, v, s, t \in \mathcal{V}$.

The degree centrality of $u$, $\mathrm{Degree}(u)$, is the most simple centrality measure and is defined to be the number of edges incident to $u$. If $\mathcal{G}$ is directed, the in-degree centrality $\mathrm{InDegree}(u)$ counts the number of incoming edges and the out-degree centrality $\mathrm{OutDegree}(u)$ is that of outgoing edges.

PageRank (Brin & Page, 1998) is a centrality measure that is most popularly known as the ranking metric of web pages. PageRank for $u$ is defined as

$$\mathrm{PR}(u) = \frac{1 - \alpha}{|\mathcal{V}|} + \alpha \sum_{v \in \mathrm{InNeighbors}(u)} \frac{\mathrm{PR}(v)}{\mathrm{OutDegree}(v)}, \tag{1}$$

where $\alpha \in [0, 1]$ is a dumping factor, $\mathrm{InNeighbors}(u)$ is the set of the nodes that have outgoing edges to $u$.

The closeness centrality's definition is

$$\mathrm{Closeness}(u) = \frac{1}{\sum_{v \in \mathcal{V}} \mathrm{Distance}(u \to v)}, \tag{2}$$

where $\mathrm{Distance}(u \to v)$ is the length of the shortest path from $u$ to $v$.

The betweenness centrality is defined as

$$\mathrm{Betweenness}(u) = \sum_{s, t \in \mathcal{V}} \frac{\sigma(s \to u \to t)}{\sigma(s \to t)}, \tag{3}$$

where $\sigma(s \to t)$ means the total number of the shortest paths from $s$ to $t$. In the similar way, $\sigma(s \to u \to t)$ means the total number of the shortest paths from $s$ to $t$ passing through $u$.

In our experiment, we used *graph-tool* (Peixoto, 2014) to calculate those centrality measures.

