# OpenReview forum: "Node Centralities and Classification Performance for Characterizing Node Embedding Algorithms"
_ICLR.cc/2018/Workshop — Reject_

### Official Review · AnonReviewer1 · 2018-03-10
**Interesting but not quite there yet**

**Rating:** 4
**Confidence:** 4

**Review:**

The idea overall is interesting: whether graph embeddings for a simple node classification task correlate with graph-wide metrics (such as centrality).  For an empirical workshop paper seeking to compare embedding methods, you would expect to find a lot more methods in the list. After asking the authors some questions, I am not convinced why other methods were not used.  The results are a bit confusing and the conclusion vague.

* GraphSAGE is basically a variant of Duvenaud that deals with the varying neighborhood sizes. The principle is the same.

PS: Please avoid keeping vital information about the experiments limited to the figure captions.

PS2: I am not as confident the results of the methods presented in the paper generalize to other methods. There is very little reason to believe that it would.

---

### Official Review · AnonReviewer4 · 2018-03-17
**Empirical comparison of node embedding**

**Rating:** 5
**Confidence:** 4

**Review:**

This paper provides an empirical comparison of some unsupervised graph node representation algorithms, by measuring classification performance in logistic regression.

It's interesting that ignoring the directed edges in the graph gives better performance, even in the methods designed for use with directed graphs. I don't think that observation alone on a couple of datasets, without further insight is quite enough.

The abstract promises that the results give insights into the properties of node embedding algorithms. However, there really isn't any analysis of what underlies the results, or why they apparently contradict previous work. As the work isn't "extremely novel", I think it needs to be a little further a long.

---

### Comment · AnonReviewer1 · 2018-03-05
**Task needs clarification**

The task described in the intro is interesting. But the node classification task is a bit unclear. There are multiple centrality measures but only one classification task. It is also unclear which metric was used in the classification task. And, why is it not a regression task? Say, predict the actual betweeness centrality metric rather than a class.

Moreover, there are a number of other node embedding methods HolE, RESCAL, Molecular fingerprints (Duvenaud et al.), GraphSAGE, GraphNN, etc.. It is unclear why we should settle for only LINE and node2vec, and what it tells us about the other methods.

Please clarify.

---

> ### Author Response · Authors · 2018-03-07
> **Response to AnonReviewer1: clarification**
>
> We thank the reviewer for finding our motivation interesting and for indicating unclear points in our manuscript. Let us clarify each of them.
>
> =================
> > But the node classification task is a bit unclear.
>
> The node classification task is to assign a class to each node in a graph. For example, in the Cora dataset, each node, a paper, is associated with one of the seven scientific fields. For other datasets please refer the last sentence of Section 2.3.
>
>
> > There are multiple centrality measures but only one classification task.
>
> Centrality measures were independent of the classification task; they were only used in the analysis stage to characterize node embedding algorithms after classification had completed. We note that node centrality measures in the analysis stage were true values, not predicted values. Moreover, Figure 1 shows only the degree centralities because, compared to other centralities, they exhibited most salient differences of embedding algorithms.
>
>
> > It is also unclear which metric was used in the classification task.
>
> We used micro F1 score averaged over the five validation folds as a node classification metric. This is written only in the caption of Table 2 due to space limitation.
>
>
> > And, why is it not a regression task? Say, predict the actual betweeness centrality metric rather than a class.
>
> We used centrality measures to understand embedding algorithms and to discuss possible enhancements; e.g. improving embeddings for low-degree nodes may be more fruitful than working for other nodes (Figure 1b). Regression, i.e. predicting centrality measures from node embeddings, is an interesting topic and in fact Rizi and Granitzer (2017) have worked on it.  Our work, however, did not involve the prediction of centrality measures; we only used centrality measures at the analysis stage to study the distributions of them on incorrectly classified nodes. Note that we used true centrality measures (not their predictions).
>
> Fatemeh Salehi Rizi and Michael Granitzer. Properties of Vector Embeddings in Social Networks. _Algorithms_, 10(4),  2017. URL http://www.mdpi.com/1999-4893/10/4/109.
>
>
> > Moreover, there are a number of other node embedding methods HolE, RESCAL, Molecular fingerprints (Duvenaud et al.), GraphSAGE, GraphNN, etc.. It is unclear why we should settle for only LINE and node2vec, and what it tells us about the other methods.
>
> We selected four algorithms, Laplacian eigenmaps, LINE-1st, LINE-2nd, and node2vec (DeepWalk), because these cover wide classes of embedding algorithms and can represent other algorithms.
> - HolE and RESCAL are similar to LINE in the sense that they minimize the distance between connected nodes; the major difference is that HolE and RESCAL are designed for knowledge graphs, which have many types of edges, whereas LINE (and we) only consider a single type of edges.
> - GraphSAGE is an extension of node2vec to calculate embeddings for new nodes that are not in training data, and we believe results obtained for node2vec should generalize fairly to GraphSAGE.
> - Molecular fingerprints are not considered a variant of the four compared algorithms but they are designed for molecules and not capable of processing wide-depth graphs such as citation or social networks used in our paper; see Section 5 in Duvenaud et al. (2015).
>
> David Duvenaud, Dougal Maclaurin, Jorge Aguilera-Iparraguirre, Rafael Gómez-Bombarelli, Timothy Hirzel, Alán Aspuru-Guzik, and Ryan P.  Adams, Convolutional Networks on Graphs for Learning Molecular Fingerprints. In _NIPS_, 2015.
>
> =================
> We thank again, and will revise the manuscript for camera-ready if accepted.

---

### Decision · Program_Chairs · 2018-03-20
**ICLR 2018 Workshop Acceptance Decision**

**Decision:**

Reject

**Comment:**

Based on the reviews, this paper has not been accepted for presentation at the ICLR workshop. However, the conversation and updates can continue to appear here on OpenReview.